# Mechanical, Flame-Retardant and Dielectric Properties of Intumescent Flame Retardant/Glass Fiber-Reinforced Polypropylene through a Novel Dispersed Distribution Mode

**DOI:** 10.3390/polym16101341

**Published:** 2024-05-09

**Authors:** Jingwen Li, Yiliang Sun, Boming Zhang, Guocheng Qi

**Affiliations:** 1School of Materials Science and Engineering, Beihang University, Beijing 100191, China; ljw666@buaa.edu.cn (J.L.); zbm@buaa.edu.cn (B.Z.); 2Department of Mechanics, Beijing Jiaotong University, Beijing 100044, China

**Keywords:** non-uniform distribution, dispersed distribution, flame retardancy, dielectric properties, IFR/GF/PP

## Abstract

The application of continuous glass fiber-reinforced polypropylene thermoplastic composites (GF/PP) is limited due to the inadequate flame retardancy of the polypropylene (PP) matrix. Apart from altering the composition of the flame retardants, the distribution modes of flame retardants also impact material performance. In this study, an alternative approach involving non-uniform distribution is proposed, namely, dispersed distribution, in which non-flame-retardant-content layers (NFRLs) and/or low-flame-retardant-content layers (LFRLs) are dispersed among high-flame-retardant-content layers (HFRLs). The mechanical, flame retardant and dielectric properties of GF/PP with intumescent flame retardant (IFR/GF/PP) are investigated comparatively under uniform, gradient, and dispersed distributions of the flame retardants. The results demonstrate that non-uniform distribution exhibits superior flame retardant performance compared to uniform distribution. Dispersed distribution enables IFR/GF/PP to attain enhanced mechanical properties and reduced dielectric constants while maintaining excellent flame-retardant properties.

## 1. Introduction

Thermoplastic composites, as a major branch of composites, have significant advantages such as rapid prototyping, repairability, secondary molding, recyclability, and environmental friendliness, in addition to light weight and high strength [1,2,3]. In continuous glass fiber (CGF)-reinforced thermoplastic composites, a PP matrix is widely used in household appliances, chemical containers, and vehicle parts due to its light weight and high strength, good wave-transparent properties, good electrical insulation properties, corrosion resistance, low price, etc. However, PP has flammable properties due to its own elemental composition and molecular structure, limiting its application [4,5,6]. Solving the problem of flame retardancy in PP often requires a large number of flame retardants, which will lead to the deterioration of PP properties, such as mechanical properties, rheological properties, and wave-transparent properties. [7]. Numerous domestic and international studies have been conducted on the flame retardancy of PP in order to obtain better comprehensive performance, and a great deal of work has been devoted to the synthesis of new flame retardants or the selection of efficient multi-component synergistic flame retardant systems [8,9,10,11,12,13,14]. However, apart from changing the flame retardants’ composition, the distribution modes of the flame retardants in the resin also affect the comprehensive performance of the materials [15,16,17].

In previous research work, there are two main ways of non-uniform distribution of flame retardants, alternating distribution and gradient distribution. Alternating distribution has HFRLs alternating with NFRLs or LFRLs at equal thickness, which is commonly found in the particle-reinforced thermoplastic resin system, and the layer-multiplying coextrusion system is used to produce the multilayered composites [18,19]. Non-combustible gases, water, and carbon layers produced by the combustion of HFRLs prevent NFRLs or LFRLs from contacting the flame. Alternating distribution multilayered composites have been shown to achieve good flame retardancy with reduced flame retardant dosage [20], improved mechanical properties [21,22], and additional features such as electromagnetic shielding [23,24], multiple shape memory effects [25,26,27], and so on. However, the alternating distribution mode is limited by the production equipment; it can generally only mold the specimen containing two flame retardant concentrations, and the thickness of each layer is similar. Thus, the layer thickness has a significant effect on the flame-retardant properties. When the layer is thin, the outer HFRLs cannot form a dense carbonization barrier of sufficient thickness on the surface, which allows the flame to spread from the outside to the inside. When the layer is thick, the carbonization barrier formed by the burning of HFRLs cannot completely cover the sections of NFRLs or LFRLs, causing flame propagation along them. Moreover, due to equipment limitations, the outermost layer always has an NFRL or LFRL on one side, which causes the flame to burn on that layer. The second type of non-uniform distribution is gradient distribution, in which the HFRLs are placed on the surface and the flame-retardant layers from the outside to the inside are HFRLs, LFRLs, and/or NFRLs [28,29,30]. Prepared by laminate molding, this preparation process allows for multiple flame retardant concentration variations and thickness variations. In gradient distribution, the outer HFRLs provide protection to the inner layer, and the thicker outer HFRLs can quickly form a dense charring barrier during the initial combustion phase, thus preventing the flame from passing to the interior. The gradient distribution mode can reduce the amount of flame retardants and improve the mechanical properties [30]. However, the gradient distribution mode appears to forget the situation of the flame burning to the inter NFRLs and/or LFRLs. Similar to the alternate distribution mode, when the NFRLs and/or LFRLs are thick, the HFRLs in the combustion process cannot form a continuous carbon layer, so that the flame cannot be effectively prevented from burning along NFRLs and/or LFRLs.

The non-uniform distribution modes of flame retardants are more reasonable and economical than the traditional uniform distribution mode. It can be inferred from the experiences of previous researchers in applying non-uniform distribution modes to resins that non-uniform distribution modes have a promising future in continuous fiber-reinforced thermoplastic composites and urgently need to be investigated. However, the above two non-uniform distribution modes have their own drawbacks, and so this article proposes another non-uniform distribution mode, dispersed distribution, in which thin LFRLs and/or NFRLs are dispersed among HFRLs. Schematic diagrams of each distribution mode are shown in Figure 1. The outermost layer is a HFRL, and the interior layers consist of HFRLs interspersed with LFRLs and/or NFRLs. Under dispersed distribution, the thickness of each layer can be different. When thick HFRLs are paired with thin NFRLs, the thicker outer HFRLs prevent heat transfer from the outside to the inside, and the dispersed thinner NFRLs can be covered by amorphous carbon produced by the combustion of internal HFRLs to prevent flame combustion along the NFRLs.

Continuous glass fiber-reinforced polypropylene composites with prepregs in sheet form can be molded by stacking the prepregs with different flame retardant concentrations, making it easier to achieve multiple and various ways of non-uniform distribution modes of flame retardants. In terms of flame retardant type, the intumescent flame retardant (IFR) is selected. During combustion, it generates carbonaceous foam layers on the surface, which have the effects of heat insulation, oxygen isolation, smoke suppression, and drip prevention. Thus, it has excellent flame-retardant properties [31,32].

In this work, IFR/GF/PP prepregs were prepared by means of melt impregnation and IFR/GF/PP laminates were prepared using the thermo compression molding process. The distribution mode of intumescent flame retardants was controlled by adjusting the arrangement order of prepregs with different flame retardant concentrations. The mechanical, dielectric, and flame-retardant properties of IFR/GF/PP were investigated under uniform, gradient, and dispersed distributions, and the impact of these distribution modes on performances was compared and analyzed.

## 2. Materials and Methods 

### 2.1. Materials

Polypropylene (PP, BX3920) with a melt flow index of 100 g/10 min (2.16 kg at 230 °C) was supplied by SK Group (Seoul, Republic of Korea). High-melt-index polypropylene (PP-MF650X) with a melt flow index of 1200 g/10 min (2.16 kg at 230 °C) was supplied by LyondellBasell Corporation (Jubail, Saudi Arabia). Maleic anhydride-grafted polypropylene (MAPP, Orevac^®^ CA100) was supplied by Arkema (Paris, France). Intumescent flame retardant (IFR), consisting of ammonium polyphosphate, pentaerythritol, and melamine, with (21 ± 1)% N and (23 ± 1)% P, was supplied by Yantai Xinxiu Corporation (Yantai, China). Direct roving of glass fiber (GF, 4305S) was supplied by the Chongqing Polycomp International Corporation (Chongqing, China).

### 2.2. Preparation of IFR/PP

PP, PP-MF650X, MAPP, and IFR were poured into the mixer according to the ratios in Table 1, and mixed well. The homogeneous mixture was poured into the twin-screw extruder for pelletizing, in order to mix the flame retardant homogeneously, and the pelletizing was repeated 3 times. Schematic diagram of preparation process is shown in Figure 2.

### 2.3. Preparation of IFR/GF/PP Prepregs

IFR/GF/PP prepregs were prepared using a continuous fiber-reinforced thermoplastic composite prepreg production line homemade in the laboratory. Flame-retardant PP pellets were melted at the extruder head position and infiltrated with continuous glass fibers, rolled, cooled, and shaped to prepare the IFR/GF/PP prepregs. The thickness of the IFR/GF/PP prepregs was 0.3 mm and the fiber content was 30 wt%. Schematic diagram of preparation process is shown in Figure 2.

### 2.4. Preparation of IFR/GF/PP Laminates

The uniform distribution of IFR/GF/PP laminates was prepared by stacking 11 layers of prepregs with the same flame retardant content. The non-uniform distribution of IFR/GF/PP laminates was prepared according to Table 2. In this paper, four uniform distribution specimens (U-0, U-10, U-20, and U-30), four gradient distribution specimens (G1, G3, G5, and G7), and five dispersed distribution specimens (D2, D3, D5, D5-10, and D5-20) were prepared, and the schematic diagrams of each specimen are shown in Figure 3. The IFR/GF/PP laminates were fabricated using the hot press molding process. The IFR/GF/PP prepregs were cut according to the size of the mold. After cutting, the prepreg sheets were arranged according to the distribution mode and placed into the mold to be heated and melted (200 °C, 20 min); then, they were quickly transferred into the molding machine at room temperature for rapid pressurization (25 °C, 7 MPa), and taken out after holding pressure for 10 min. Finally, they were cut into the required sizes for each test using waterjet cutting equipment. In order to reduce the effect of fiber orientation on performance, the IFR/GF/PP laminates were unidirectional. Schematic diagram of preparation process is shown in Figure 2.

### 2.5. Characterization

#### 2.5.1. Mechanical Properties

Mechanical property tests were performed on Changchun Kexin WDW-100 Universal Mechanical Testing Machine. According to the ASTM D7264 standard [33], a three-point bending performance test was carried out on each specimen at the beam moving speed of 2 mm/min. The span-to-thickness ratio was 32, meaning that the span was 32 × 3.3 = 105.6 mm. There were 5 parallel specimens for each sample. The bending strength and bending modulus in the text were the average values of the 5 parallel specimens, and the error bars were the standard deviations.

#### 2.5.2. Dielectric Properties

The dielectric constant and dielectric loss of each specimen at X-band (8.2–12.4 GHz) electromagnetic waves were measured using the rectangular waveguide method on a Model AV3672C Vector Grid Analyzer (China Electronics Technology Group Corporation, Qingdao, China). The rectangular waveguide dielectric constant measurement method utilized the TE mode of electromagnetic waves transmitted from the waveguide to the material to be measured, and the amplitude and phase information of the reflected and transmitted signals were used to invert the dielectric constant of the material to be measured. The specimen size was 10.16 mm × 22.86 mm × 3.3 mm. Because the dielectric properties are related to the fiber orientation (Section 3.1 for the theory), two fiber orientations were chosen and two parallel specimens for each orientation, and the data in the text are the average values of the two parallel specimens.

#### 2.5.3. Limiting Oxygen Index (LOI)

The Limiting Oxygen Index of each specimen was measured using an Oxygen Indexer (Nanjing Jiangning Analytical Instrument Co., Ltd., Nanjing, China), according to the ASTM D2863 standard [34] test method. The LOI measured the minimum concentration of oxygen required for the flaming combustion of the material in a mixed oxygen-nitrogen gas stream. The LOI specimen size was 6.5 mm × 120 mm × 3.3 mm. The number of parallel specimens was about 15, and the experimental results were statistical results according to ASTM D2863.

#### 2.5.4. Vertical Burning Test

Flame retardant rating was tested using a vertical combustion testing machine (YK-Y0142, YAOKE, Nanjing, China) according to the UL94 flame retardant rating (FRR) test method. The flame retardancy rating was used to evaluate the ability of a material to extinguish after ignition. The FRR specimen size was 13 mm × 120 mm × 3.3 mm. The number of parallel specimens was 10, and the experimental results were statistical results according to UL94.

#### 2.5.5. Cone Calorimeter Test (CCT)

A cone calorimeter test was carried out on each specimen using the cone calorimeter (Fire Testing Technology, Leeds, UK) with a heat flow of 35 kW/m^2^ according to ISO 5660 [35]. The CCT provided many parameters of the combustion of a combustible material in a fire, including heat release rate (HRR), total heat release (THR), effective heat of combustion (EHC), time to ignition (TTI), smoke and toxicity parameters, mass change parameters (MLR), etc. The CCT specimen size was 100 mm × 100 mm × 3.3 mm.

The fiber orientation of the specimens in each test is shown in Figure 4.

## 3. Results and Discussion

### 3.1. Uniform Distribution

Under the uniform distribution mode, the flame retardant content in each layer of prepregs of IFR/GF/PP is the same. Figure 5 shows the bending properties, dielectric properties, and flame-retardant properties of each specimen in the uniform distribution mode.

The bending strength and bending modulus of IFR/GF/PP show a decreasing trend with the increase in flame retardant content. When the flame retardant content is low, the flame retardant has a low effect on the interface and mainly plays the role of enhancing the mechanical properties of the resin matrix, and the overall bending strength and bending modulus of the composites do not decrease or even slightly increase due to the small amount of flame retardant added. When a large amount of flame retardant is added, the resin melt viscosity increases, fluidity decreases, and the resin’s infiltration of the fiber decreases, resulting in a reduction in the interfacial properties between the resin and the fiber. Moreover, the flame retardant causes imperfections in the resin, which ultimately results in a reduction in the bending properties of IFR/GF/PP. The bending strength and bending modulus of U-30 are decreased by 18.07% and 22.86%, respectively, compared with those of U-0.

The dielectric constant and dielectric loss tangent of IFR/GF/PP reduce with the increase in electromagnetic wave frequency and rise with the increase in flame retardant content. Since the dielectric constant and dielectric loss tangent of flame retardants are higher than that of PP, and the interfacial polarization between resin and fiber increases with the increase in flame retardant content, this improves the dielectric constant and dielectric loss tangent of IFR/GF/PP. The dielectric properties of unidirectional continuous fiber composite laminates decrease with the growth of the angle θ (Figure 4b) between the fibers and the direction of the electric field, with a maximum at θ_1_ = 0°(parallel) (Figure 4c) and a minimum at θ_2_ = 90°(vertical) (Figure 4d). The relation between the dielectric constant and the angle θ of the continuous fiber-reinforced composite [36] is shown in Equation (4). This rule does not change with the variation in flame retardant content.
(1)$$\varepsilon_{11}^{\prime }=\varepsilon_{f}^{\prime }\upsilon_{f}+\varepsilon_{m}^{\prime }\upsilon_{m}$$
(2)$$\varepsilon_{22}^{\prime }=\frac{\varepsilon_{m}^{\prime 2}+\varepsilon_{m}^{\prime }\sqrt{\upsilon_{f}}\left({\bar{\varepsilon }}_{f2}^{\prime }-\varepsilon_{m}^{\prime }\right)}{\varepsilon_{m}^{\prime }\left(1+\upsilon_{f}-\sqrt{\upsilon_{f}}\right)+{\bar{\varepsilon }}_{f2}^{\prime }\left(\sqrt{\upsilon_{f}}-\upsilon_{f}\right)}$$
where
(3)$${\bar{\varepsilon }}_{f2}^{\prime }=\varepsilon_{m}^{\prime }+\frac{\pi }{4}\left(\varepsilon_{f}^{\prime }-\varepsilon_{m}^{\prime }\right)$$
(4)$${\left[\varepsilon_{ij}^{\prime }\right]}^{\theta }=\varepsilon_{11}^{\prime }{cos}^{2}\theta +\varepsilon_{22}^{\prime }{sin}^{2}\theta$$
where $\varepsilon_{11}^{\prime }$ and $\varepsilon_{22}^{\prime }$ are the dielectric constants at angles of 0 and 90, respectively. $\varepsilon_{f}^{\prime }$ and $\varepsilon_{m}^{\prime }$ are the dielectric constants of the fiber and matrix, respectively. $\upsilon_{f}$ and $\upsilon_{m}$ are the volume contents of the fiber and matrix, respectively, and ${\left[\varepsilon_{ij}^{\prime }\right]}^{\theta }$ is the dielectric constant when the angle is θ.

The flame retardant rating and LOI of IFR/GF/PP are enhanced with the increase in flame retardant content. The LOI of U-30 reaches 29.0%, and the flame retardant rating of U-30 reaches V0. As the most important role of flame retardant, the increase in flame retardant obviously improves the flame-retardant property of IFR/GF/PP. IFR can form a layer of amorphous carbon covering the surface of materials when it burns, and such a porous carbon layer can effectively isolate the source of ignition and slow down the spread of the flame, thus preventing the materials from continuing to be attacked by the flame.

### 3.2. Gradient Distribution

In gradient distribution, HFRLs are placed on the outside and the flame retardant concentration decreases from the outside to the inside. Figure 6 shows the bending, dielectric, and flame-retardant properties of each specimen under the gradient distribution mode.

It is expected that increasing the proportion of NFRL in composite laminates can improve the mechanical properties. Figure 6a,b gives the results as expected: when the NFRLs are concentrated inside the structure, the mechanical properties of the composite laminates all increase with the increase in the percentage of NFRLs. The bending strength and bending modulus of G7 are enhanced by 11.52% and 18.28%, respectively, compared with those of U-30.

The dielectric constant and dielectric loss tangent of the IFR/GF/PP show a decreasing trend with the increase in frequency, and do not vary with the change in flame retardant content. The addition of NFRLs leads to the reduction in flame retardant content in IFR/GF/PP, and so the dielectric constant and dielectric loss tangent decrease with the increase in the proportion of NFRLs.

The LOI and flame retardant rating of IFR/GF/PP decrease with the increase in the proportion of NFRLs in gradient distribution. The outer HFRLs burn to form an expansive carbon layer, which attempts to cover the surface of NFRLs and prevent the dripping of molten resin. The combustion of IFR forms water vapor and flame retardant gases, which insulate the air and ultimately prevent the internal non-flame-retardant layer from continuing to burn. The NFRLs are flammable, and once exposed to flame in the cross-section, the internal NFRLs, playing the role of “candle wick”, will be the first to melt and ignite, and then the flame will burn along the NFRLs to the outside. When the number of NFRLs is smaller, LOI decreases slowly, but it decreases rapidly when the number of NFRLs increases from 5 to 7. This is due to the fact that after the outer HFRLs are thinned to a certain thickness, it becomes difficult for the expanded carbon from the combustion to completely cover the inner NFRLs. In addition, the thinned outer HFRLs have weak deformation resistance, and combustion causes them to expand and bend outward, exposing more NFRLs. In this situation, the outer HFRLs do not provide good protection for the inner NFRLs, and the flame continues to burn along the NFRLs.

### 3.3. Dispersed Distribution

In the dispersed distribution mode, LFRLs and/or NFRLs are distributed among HFRLs. Figure 7 shows the bending properties, dielectric properties, and flame-retardant properties of the specimens in the dispersed distribution mode.

Similar to gradient distribution, the mechanical properties of IFR/GF/PP are enhanced with the increase in the number of NFRLs under the dispersed distribution mode. The bending strength and bending modulus of D5 are increased by 8.05% and 10.85%, respectively, compared with those of U-30. The properties of IFR/GF/PP are positively correlated with those of the LFRLs when the LFRLs have the same distribution location and number of layers as the NFRLs, and only the flame retardant concentration is changed. As a result, the mechanical properties of D5-10 are slightly improved compared with those of D5, but the mechanical properties of D5-20 are decreased when the flame retardant concentration continues to increase.

The increase in the number of flame-retardant-free layers leads to a reduction in the flame retardant content and consequently to a decrease in both the dielectric constant and dielectric loss tangent. The dielectric constant and dielectric loss tangent decrease with the reduction in flame retardant concentration in LFRLs.

The flame retardancy of the composite laminates in the dispersed distribution mode declines with the increase in the number of NFRLs and is enhanced with the increase in the flame retardant concentration in the LFRLs. Significantly, it is easier to obtain high flame retardant ratings in the dispersed distribution mode. The flame retardant rating of D2 and D5-20 reaches V0, and the flame retardant rating of D3 and D5-10 reaches V1.

### 3.4. Performance Comparison by Distribution Mode

Figure 8 shows the mechanical, dielectric, and flame-retardant properties of IFR/GF/PP with different distribution modes.

Figure 8a shows that except for the LFRL dispersed distribution, the bending strength and bending modulus of each specimen in other distribution modes decreased with the increase in flame retardant content. The dispersed distribution of LFRLs changes the concentration of the LFRL, causing the mechanical properties of IFR/GF/PP to change according to the mechanical properties of the LFRLs. Therefore, D-10 with IFR-10 as the LFRL has a better bending strength and modulus than D with IFR-0 and D-20 with IFR-20. At the same flame retardant content of composite laminates, the mechanical properties of IFR/GF/PP in the uniform distribution mode are better than those in the non-uniform distribution mode, except for the LFRL dispersed distribution. The mechanical properties of dispersed distribution with LFRLs are higher than those of the uniform distribution mode at a higher flame retardant content, whereas the gradient distribution pattern has the worst mechanical properties. Comparing G3 and D3, G5 and D5 with the same flame retardant content in the gradient and dispersed distribution modes, it is found that the composite laminates with the dispersed distribution mode have a higher bending strength and bending modulus. This is due to the dispersed distribution mode in which NFRLs or LFRLs are distributed closer to the surface of the laminates, so that the composite plate forms a more approximate sandwich structure with strong outside, and when the composite plate is subjected to the bending load, the upper layers are able to withstand the compressive stress. This can be explained by the relevant knowledge of mechanics. The relationship between the bending modulus of the laminate and the modulus and position of each layer of the laminate is as follows:(5)$$E_{P}=\frac{8}{h^{3}}\sum_{i=0}^{N/2}E_{i}(z_{i}^{3}-z_{i-1}^{3})$$
where *E_P_* is the bending modulus of the laminate, *E_i_* is the modulus of layer *i*, *i* is the number of plies counted outward from the neutral plane, *h* is the total thickness of the laminate, *N* is the total number of plies, and *z* is the distance from the *i*th ply to the neutral plane, as shown in Figure 9.

The flexural modulus of G3 and D3 is calculated according to Equation (5):(6)$$\begin{array}{l} E_{G3}=\frac{8}{{\left(11t\right)}^{3}}\left(E_{IFR\text{-}0}\left({\left(\frac{3}{2}t\right)}^{3}-0^{3}\right)+E_{IFR\text{-}30}\left({\left(\frac{11}{2}t\right)}^{3}-{\left(\frac{3}{2}t\right)}^{3}\right)\right)\\ =\frac{3^{3}}{11^{3}}E_{IFR\text{-}0}+\frac{11^{3}-3^{3}}{11^{3}}E_{IFR\text{-}30}\\ \approx 0.02E_{IFR\text{-}0}+0.98E_{IFR\text{-}30} \end{array}$$
(7)$$\begin{array}{l} E_{D3}=\frac{8}{{\left(11t\right)}^{3}}\left(E_{IFR\text{-}0}\left({\left(\frac{1}{2}t\right)}^{3}-0^{3}\right)+E_{IFR\text{-}30}\left({\left(\frac{5}{2}t\right)}^{3}-{\left(\frac{1}{2}t\right)}^{3}\right)+E_{IFR\text{-}0}\left({\left(\frac{7}{2}t\right)}^{3}-{\left(\frac{5}{2}t\right)}^{3}\right)+E_{IFR\text{-}30}\left({\left(\frac{11}{2}t\right)}^{3}-{\left(\frac{7}{2}t\right)}^{3}\right)\right)\\ =\frac{3^{3}}{11^{3}}E_{IFR\text{-}0}+\frac{5^{3}-1}{11^{3}}E_{IFR\text{-}30}+\frac{7^{3}-5^{3}}{11^{3}}E_{IFR\text{-}0}+\frac{11^{3}-7^{3}}{11^{3}}E_{IFR\text{-}30}\\ \approx 0.16E_{IFR\text{-}0}+0.84E_{IFR\text{-}30} \end{array}$$
where *E_G_*_3_ and *E_D_*_3_ are the bending modulus of G3 and D3. *E_IFR_*_-0_ and *E_IFR_*_-30_ are the modulus of IFR-0 and IFR-30, and *t* is the thickness of each layer.

Based on the results of Equations (6) and (7), it is observed that the flexural modulus of D3 is higher than that of G3. Hence, when the NFRLs or LFRLs with high mechanical properties are closer to the surface layers, the composite laminates can resist stronger bending deformation and withstand higher bending loads. Composite laminates with better mechanical properties of the outer layer under dispersed distribution result in higher bending properties.

Figure 8c,d show that in each distribution mode, the dielectric constant and dielectric loss tangent increase with the increase in flame retardant content, and tend to decrease with the increase in frequency. Specimens G3 and D3, as well as G5 and D5, from gradient and dispersed distributions, respectively, which contain the same flame retardant concentration, obtain similar dielectric properties.

The IFR is dispersed as particles within the matrix, and so the dielectric constant of the matrix can be calculated using Equation (8).
(8)$$\varepsilon_{m}^{\prime }=v_{pp}\varepsilon_{pp}^{\prime }+v_{IFR}\varepsilon_{IFR}^{\prime }$$

Bringing Equation (8) into Equation (1), the formula for the dielectric constant of IFR/GF/PP with single flame retardant content in the parallel direction (θ_1_ = 0°) can be obtained:(9)$$\varepsilon_{11}^{\prime }=v_{f}\varepsilon_{f}^{\prime }+v_{pp}\varepsilon_{pp}^{\prime }+v_{IFR}\varepsilon_{IFR}^{\prime }$$
where $v_{f}+v_{pp}+v_{IFR}=1$, and so Equation (9) can be written as Equation (10):(10)$$\varepsilon_{11}^{\prime }=v_{f}\varepsilon_{f}^{\prime }+(1-v_{f}-v_{IFR})\varepsilon_{pp}^{\prime }+v_{IFR}\varepsilon_{IFR}^{\prime }$$

Since the volume content of fibers in the prepreg used in this paper is constant and the dielectric constant of IFR is higher than that of PP, according to Equation (10), it can be concluded that the dielectric constant of uniform-distribution IFR/GF/PP increases with the increase in flame retardant content.

The dielectric properties of non-uniform-distribution IFR/GF/PP are related to the dielectric properties of each prepreg layer:(11)$$\varepsilon_{p}^{\prime }=\frac{1}{h}\sum_{i=1}^{N}t_{i}\varepsilon_{i}^{\prime }$$

Because the thickness of each layer is the same, Equation (11) can be written as:(12)$$\varepsilon_{p}^{\prime }=\frac{N_{0}}{N}\varepsilon_{0}^{\prime }+\frac{N_{10}}{N}\varepsilon_{10}^{\prime }+\frac{N_{20}}{N}\varepsilon_{20}^{\prime }+\frac{N_{30}}{N}\varepsilon_{30}^{\prime }$$
where $\varepsilon_{P}^{\prime }$ is the dielectric constant of the IFR/GF/PP laminate, *N* is the total number of layers, and *N*_0_, *N*_10_, *N*_20_, and *N*_30_ are the number of layers of IFR-0, IFR-10, IFR-20, and IFR-30, respectively.

Ultimately, the dielectric constant of the laminate is:(13)$$\varepsilon_{p}^{\prime }=v_{f}\varepsilon_{f}^{\prime }+\left(1-v_{f}-v_{IFR}^{\prime }\right)\varepsilon_{pp}^{\prime }+v_{IFR}^{\prime }\varepsilon_{IFR}^{\prime }$$
where $v_{IFR}^{\prime }$ is the volume fraction of IFR to the whole of the laminate; Equation (13) contains Equation (10).

It can be concluded that the dielectric properties of IFR/GF/PP have a less strong relationship with the distribution modes, but are mainly related to the overall flame retardant content of IFR/GF/PP laminates, and tend to rise with the increase in flame retardant content.

Figure 8b shows that the flame-retardant properties under each distribution mode improve with the increase in flame retardant content. The LOI is highest in the gradient distribution mode at a lower flame retardant content, followed by dispersed distribution, and lowest in uniform distribution, whereas when the flame retardant content is higher, such advantage is no longer significant. The reason is that the flame-retardant properties of prepregs increase slowly with the increase in flame retardant content, and only after the flame retardant content reaches a certain value can the flame retardant properties be significantly improved. When the overall flame retardant content of the composite laminate is low, the flame-retardant properties of the HFRLs in the non-uniform distribution mode have remarkable advantages, which improve the difficulty of combustion and resist the spread of flame. When the flame retardant content is high, the flame-retardant properties of the HFRLs in the non-uniform distribution mode are weakened, and the LFTLs act as “candle wick”, which further reduces the flame-retardant properties. Comparing gradient distribution and dispersed distribution, dispersed distribution has a higher flame retardant rating under the same flame retardant content. Gradient distribution fails to significantly improve the flame retardant rating of the composite laminates, but the gradient distribution mode has a slightly higher LOI. It is mainly due to the different combustion methods of the LOI and the flame retardant rating test. The LOI test method is to ignite the upper end of the sample and burn it downwards, while the flame retardant rating test method is to ignite the lower end of the sample and burn it upwards. The molten resin matrix by gravity flows towards the unburned portion of the composite laminate in the LOI test and poses no threat, whereas in the flame retardant rating test, the situation is exactly the opposite, as the molten droplets flow towards the flame and become a fuel for combustion, which aggravates the combustion of the composite laminate. Under dispersed distribution, the contact area between LFRLs and HFRLs or NFRLs and HFRLs is significantly increased, and the expanding carbon formed by the HFRLs after combustion forms grooves in these contact positions, hindering the melt flow of the resin matrix and preventing the resin droplets of the NFRLs or LFRLs from contacting the flame, thus slowing down the combustion to prevent the flame from spreading and improving the flame retardant rating (Figure 10). Under the gradient distribution mode, the outermost layer consists of HFRLs, and the thicker HFRLs have a higher LOI. When the outermost layer is not burned, it is difficult to get continuous oxygen support for the inner layer to burn. Meanwhile, at the initial combustion stage, the thicker HFRLs can produce abundant expanding carbon layer covering the surface of the inner layer of NFRLs or LFRLs, which prevents the inner layers from contacting the oxygen, and so composites under the gradient distribution mode have a slightly higher LOI. However, the LOI specimens of gradient distribution are difficult to self-extinguish once they burn up. This is due to the fibers that do not burn and have a large amount of expanded carbon attached to them, bending and deforming to the outside by gravity; the deformation directly exposes the internal NFRLs or LFRLs to oxygen, causing the flame to burn continuously along the internal NFRLs or LFRLs and making it difficult to self-extinguish. The digital photographs of a gradient distribution sample and a dispersed distribution sample during combustion are shown in Figure 11.

Using CCT, the fire hazard comparison analysis of three IFR/GF/PP specimens (G1, D5-20, and D2) with flame retardant rating up to V0 is carried out. Figure 12a,b show the thermal hazards of the samples: the HRR curve and the THR curve. As well as thermal hazards, fire non-thermal hazards are also very important, such as whether the product is toxic or corrosive, or smoke production. Consequently, the smoke production rate (SPR, Figure 12c) and the amount of smoke production (TSP, Figure 12d) for G1, D5-20, and D2 are discussed. The TTI, peak HRR (PHRR), time to peak HRR (TPHRR), THR, and fire performance index (FPI = TTI/PHRR) of each sample at a heat flow density of 35 kW/m^2^ are listed in Table 3. The flame retardant contents of G1, D5-20, and D2 are 18.81%, 18.02%, and 16.93%, respectively. The TTI of combustible materials is the time from exposure to a thermal radiation source to sustained ignition on the surface of the sample. It is an important parameter of fire hazard, and materials that are more prone to ignition have a higher fire hazard. The ignition time of the three specimens is the same, namely, 30 s, and they all have good fire resistance performance. The decrease in flame retardant content under non-uniform distribution did not reduce the ignition time. This indicates that the outer HFRLs of D5-20 and D2 effectively block the heat. The HRR is the amount of heat released per unit time and per unit area of the sample when it burns, which is an important parameter to measure the degree of fire hazard. The PHRRs of G1, D5-20, and D2 are 121.64 kw/m^2^, 130.58 kw/m^2^, and 137.15 kw/m^2^, respectively, which increase slightly with the decrease in flame retardant content. The FPI is defined by the ratio of the TTI value to the PHRR value, which is chosen to further evaluate the fire hazard of the samples, and the larger the FPI value, the lower the fire hazard. The FPI values of G1, D5-20, and D2 are 0.2466 s.m^2^/kw, 0.2187 s.m^2^/kw, and 0.2297 s.m^2^/kw, respectively, which decrease slightly with the reduction in flame retardant content. The thermal hazards of G1, D5-20, and D2 are generally slightly increased with the decrease in flame retardant content. But Figure 12c,d show that the SPR and TSP of D2, which has the lowest flame retardant content, are significantly lower than those of D5-20. Compared with D5-20, D2 has a thicker outer HFRL. The thicker outer HFRL produces thicker expanding carbon layer when it burns, which effectively reduces the generation of smoke.

## 4. Conclusions

The uniform distribution mode of a large amount of flame retardant in GF/PP can significantly improve its flame-retardant properties, but this results in reducing the mechanical properties and increasing the dielectric properties. At the same flame retardancy level, the non-uniform distribution pattern of flame retardant can enable materials to have better mechanical properties and lower dielectric constants with a smaller amount of flame retardant. The non-uniform distribution mode of flame retardant can better balance the relationship among mechanical, dielectric, and flame-retardant properties.

Under the same flame retardant content, uniform distribution has higher mechanical properties than non-uniform distribution, but in this situation, uniform distribution has poor flame-retardant properties. In non-uniform distribution, dispersed distribution, in which the low-flame-retardant layer with better mechanical properties is distributed closer to the surface layer, has higher mechanical properties than gradient distribution.

The dielectric properties are positively correlated with the flame retardant content. The dielectric constant and dielectric loss tangent of IFR/GF/PP can be reduced by decreasing the flame retardant dosage through non-uniform distribution.

At the same flame retardant content, non-uniform distribution has better flame-retardant properties than uniform distribution, especially at lower overall flame retardant content. At the same flame retardant content, comparing gradient distribution and dispersed distribution, gradient distribution has a slightly higher LOI, and dispersed distribution has a higher flame retardant grade, caused by the gravity of the fiber and the molten resin droplets during the combustion process.

According to the results of the cone calorimeter test, the expanding carbon layer generated by the combustion of a thicker outer flame-retardant layer can effectively block heat and reduce the generation of smoke.

In dispersed distribution, IFR/GF/PP laminates have excellent flame-retardant properties in spite of the reduced flame retardant content. The dispersed distribution mode satisfies the requirement that the outer HFRLs prevent heat transfer from the outside to the inside, and the internal NFRLs or LFRLs can be covered by the amorphous carbon produced by the combustion of the internal HFRLs on both sides to prevent flames from burning along the NFRLs or LFRLs. And because the NFRLs or LFRLs with high mechanical properties are closer to the surface layers, the composite laminates have better bending properties. Hence, dispersed distribution is more suitable for applications subject to bending loads. Dispersed distribution can enable IFR/GF/PP to obtain better mechanical properties and a lower dielectric constant, satisfying the flame retardant performance and making IFR/GF/PP suitable for more application scenarios.

## Figures and Tables

**Figure 1 polymers-16-01341-f001:**
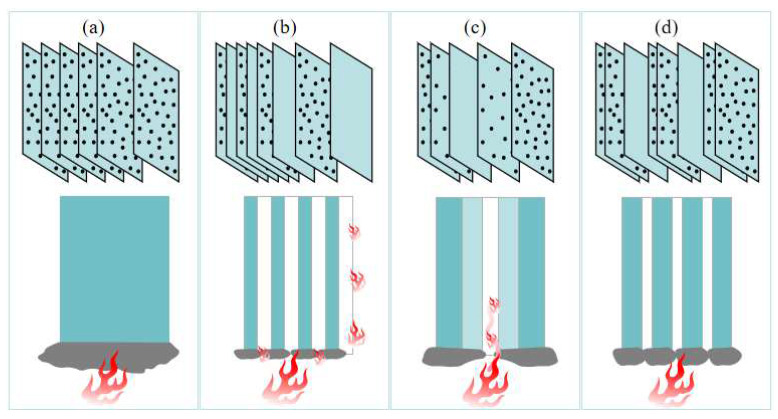
Schematic diagrams of each distribution mode: (**a**) uniform distribution, (**b**) alternating distribution, (**c**) gradient distribution, and (**d**) dispersed distribution.

**Figure 2 polymers-16-01341-f002:**
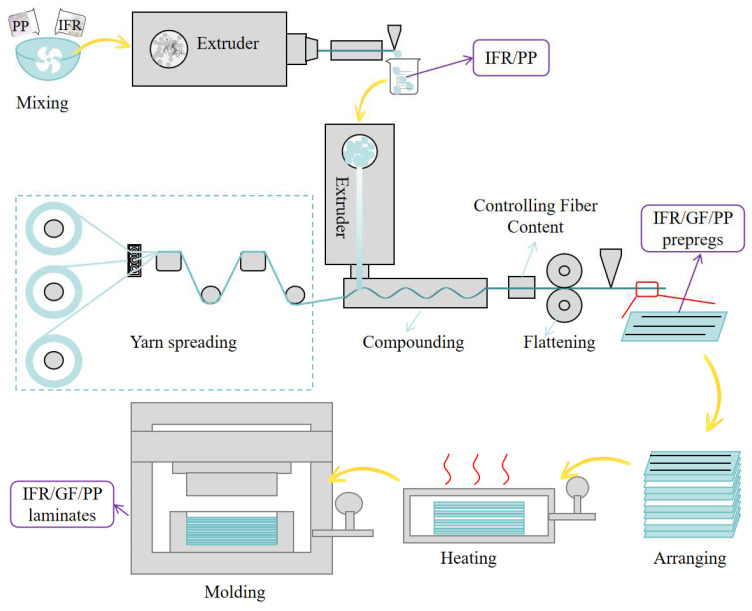
Schematic diagram of preparation process.

**Figure 3 polymers-16-01341-f003:**
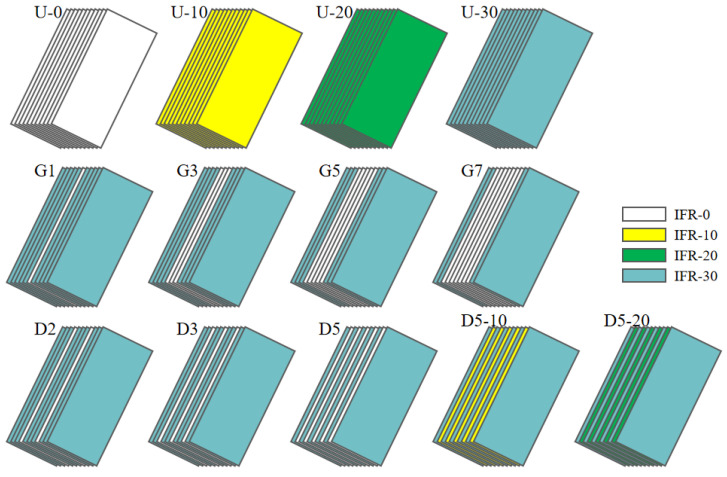
Schematic diagrams of IFR/GF/PP laminates.

**Figure 4 polymers-16-01341-f004:**
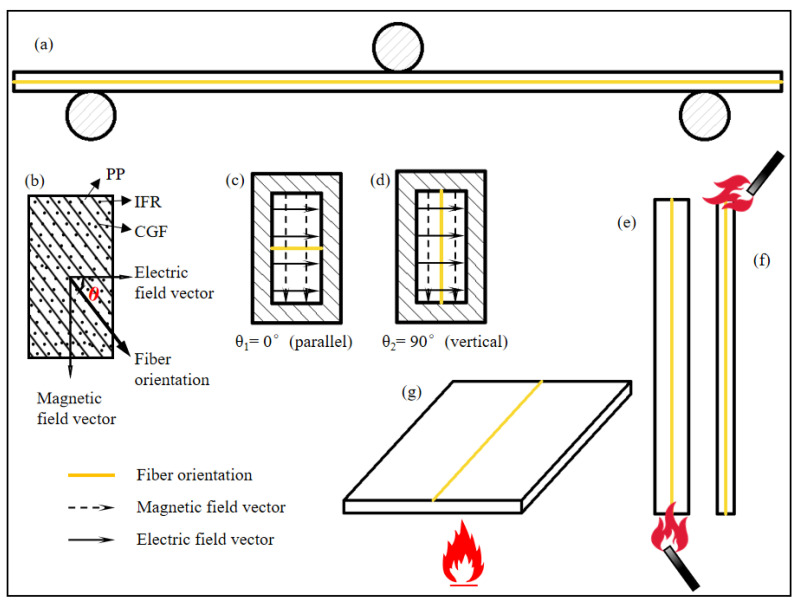
Fiber orientation of the specimens in each test: (**a**) bending property test, (**b**–**d**) dielectric property test, (**e**) FRR, (**f**) LOI, and (**g**) CCT.

**Figure 5 polymers-16-01341-f005:**
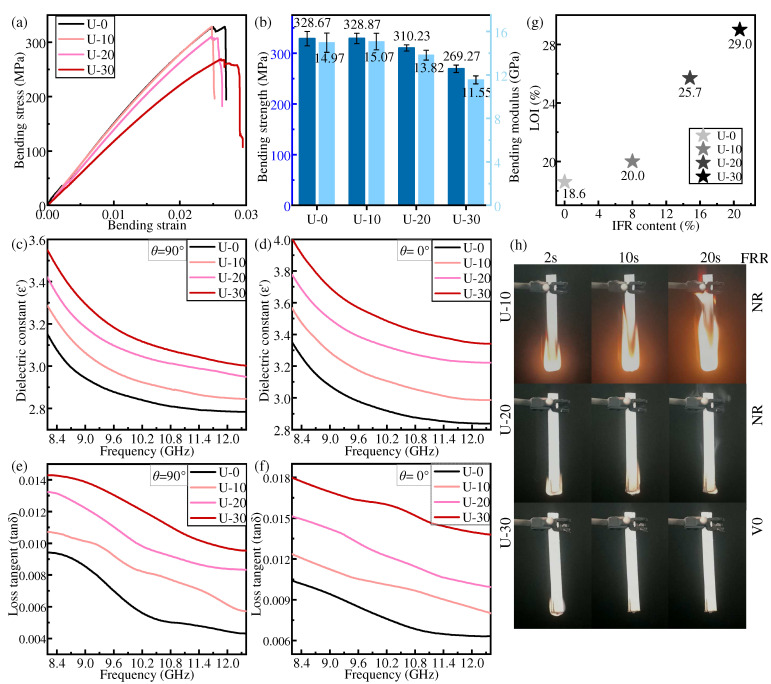
Properties of samples in uniform distribution: (**a**,**b**) bending property, (**c**,**d**) dielectric constant, (**e**,**f**) dielectric loss tangent, (**g**) LOI, and (**h**) FRR.

**Figure 6 polymers-16-01341-f006:**
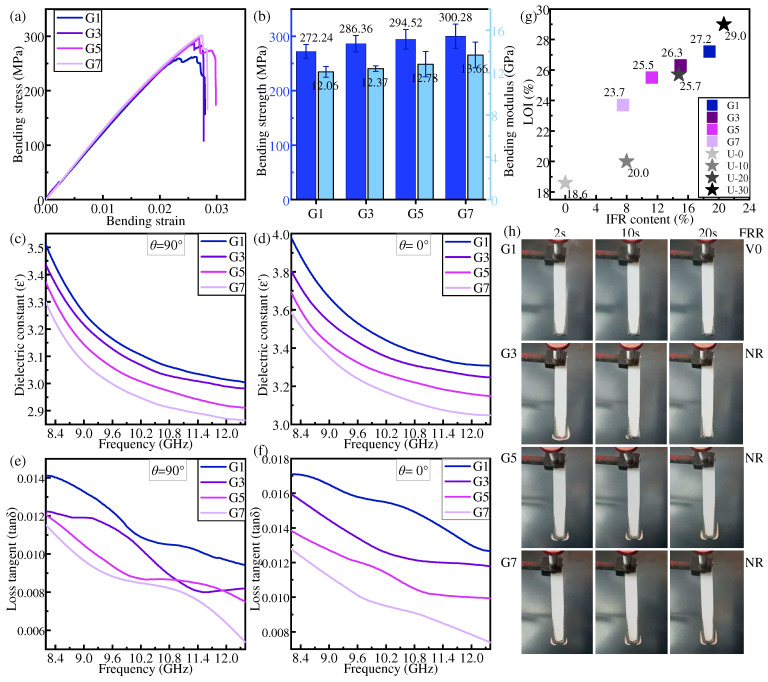
Properties of samples in gradient distribution: (**a**,**b**) bending property, (**c**,**d**) dielectric constant, (**e**,**f**) dielectric loss tangent, (**g**) LOI, and (**h**) FRR.

**Figure 7 polymers-16-01341-f007:**
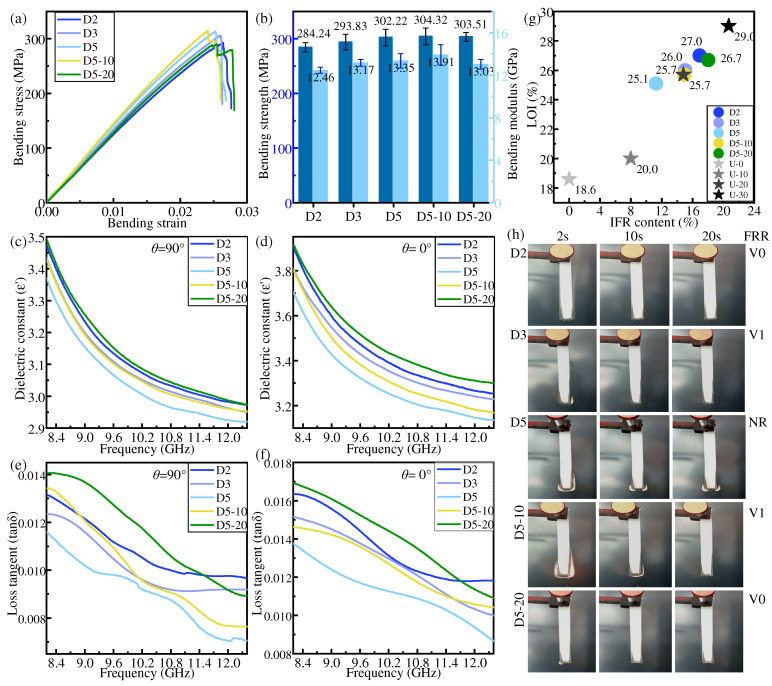
Properties of samples in dispersed distribution: (**a**,**b**) bending property, (**c**,**d**) dielectric constant, (**e**,**f**) dielectric loss tangent, (**g**) LOI, (**h**) and FRR.

**Figure 8 polymers-16-01341-f008:**
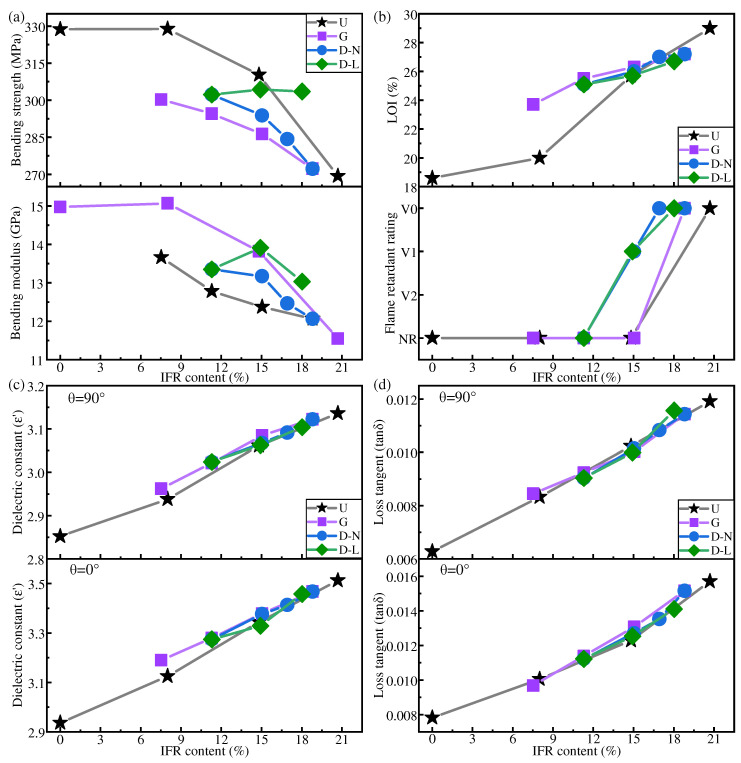
Mechanical properties (**a**), flame-retardant properties (**b**), dielectric constant at 10 GHz (**c**), and average value of dielectric loss tangent (**d**) of IFR/GF/PP with different distribution modes (U: uniform distribution. G: gradient distribution. D-N: dispersed distribution with NFRLs. D-L: dispersed distribution with LFRLs).

**Figure 9 polymers-16-01341-f009:**
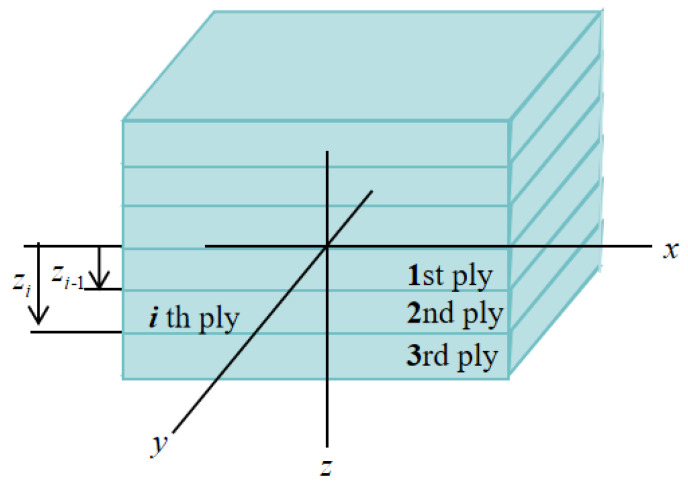
Schematic diagram of the laminate.

**Figure 10 polymers-16-01341-f010:**
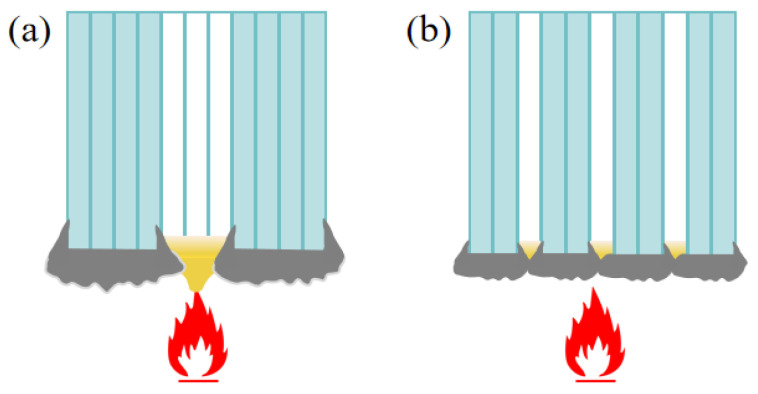
Schematic diagrams of samples during combustion: (**a**) gradient distribution sample and (**b**) dispersed distribution sample.

**Figure 11 polymers-16-01341-f011:**
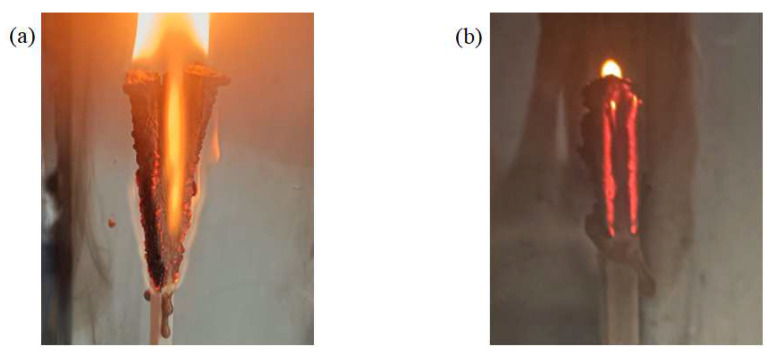
Digital photographs of samples during LOI test: (**a**) gradient distribution sample and (**b**) dispersed distribution sample.

**Figure 12 polymers-16-01341-f012:**
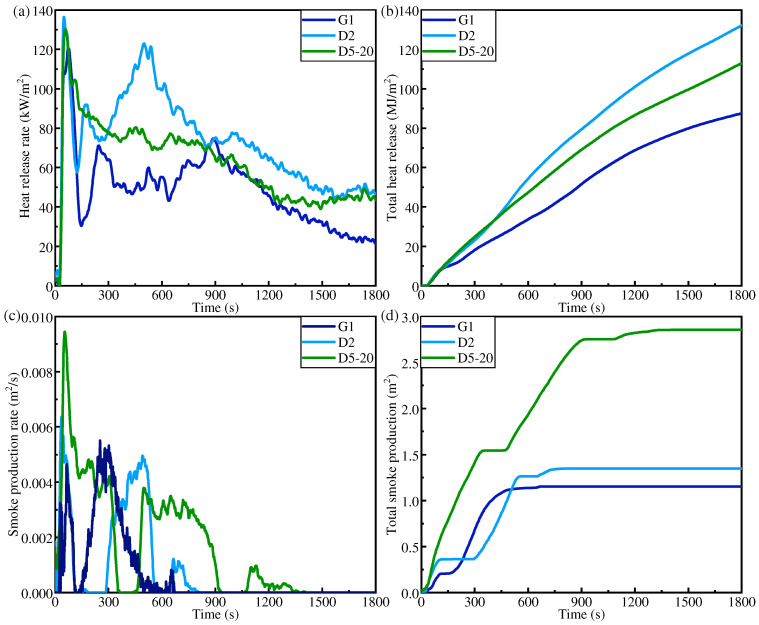
CCT results of G1, D2, and D5-20: (**a**) HRR, (**b**) THR, (**c**) SPR, (**d**) TSP.

**Table 1 polymers-16-01341-t001:** Designation and composition of IFR/PP.

Samples	PP (phr)	MAPP (phr)	PP-MF650X (phr)	IFR (phr)	IFR Content (wt%)
IFR-0	100	5	10	0	0
IFR-10	100	5	10	10	8.00
IFR-20	100	5	10	20	14.81
IFR-30	100	5	10	30	20.69

IFR content = IFR/(PP + MAPP + PP − MF650X + IFR).

**Table 2 polymers-16-01341-t002:** Resin matrix composition and distribution in each layer of IFR/GF/PP laminates.

Samples	Resin Matrix Composition and Distribution in Each Prepreg Layer											IFR Content (wt%)
U-0	IFR-0	IFR-0	IFR-0	IFR-0	IFR-0	IFR-0	IFR-0	IFR-0	IFR-0	IFR-0	IFR-0	0
U-10	IFR-10	IFR-10	IFR-10	IFR-10	IFR-10	IFR-10	IFR-10	IFR-10	IFR-10	IFR-10	IFR-10	8.00
U-20	IFR-20	IFR-20	IFR-20	IFR-20	IFR-20	IFR-20	IFR-20	IFR-20	IFR-20	IFR-20	IFR-20	14.81
U-30	IFR-30	IFR-30	IFR-30	IFR-30	IFR-30	IFR-30	IFR-30	IFR-30	IFR-30	IFR-30	IFR-30	20.69
G1	IFR-30	IFR-30	IFR-30	IFR-30	IFR-30	IFR-0	IFR-30	IFR-30	IFR-30	IFR-30	IFR-30	18.81
G3	IFR-30	IFR-30	IFR-30	IFR-30	IFR-0	IFR-0	IFR-0	IFR-30	IFR-30	IFR-30	IFR-30	15.05
G5	IFR-30	IFR-30	IFR-30	IFR-0	IFR-0	IFR-0	IFR-0	IFR-0	IFR-30	IFR-30	IFR-30	11.29
G7	IFR-30	IFR-30	IFR-0	IFR-0	IFR-0	IFR-0	IFR-0	IFR-0	IFR-0	IFR-30	IFR-30	7.52
D2	IFR-30	IFR-30	IFR-30	IFR-0	IFR-30	IFR-30	IFR-30	IFR-0	IFR-30	IFR-30	IFR-30	16.93
D3	IFR-30	IFR-30	IFR-0	IFR-30	IFR-30	IFR-0	IFR-30	IFR-30	IFR-0	IFR-30	IFR-30	15.05
D5	IFR-30	IFR-0	IFR-30	IFR-0	IFR-30	IFR-0	IFR-30	IFR-0	IFR-30	IFR-0	IFR-30	11.29
D5-10	IFR-30	IFR-10	IFR-30	IFR-10	IFR-30	IFR-10	IFR-30	IFR-10	IFR-30	IFR-10	IFR-30	14.92
D5-20	IFR-30	IFR-20	IFR-30	IFR-20	IFR-30	IFR-20	IFR-30	IFR-20	IFR-30	IFR-20	IFR-30	18.02

The IFR content in this article is calculated as the overall flame-retardant-mass content in the resin matrix of the composite laminate; the fiber mass is not included.

**Table 3 polymers-16-01341-t003:** CCT results of G1, D2, and D5-20.

Samples	IFR Content (wt%)	TTI (s)	TPHRR (s)	PHRR (kw/m^2^)	THR (MJ/m^2^)	FPI (s.m^2^/kW)
G1	18.81	28	74	121.64	88.23	0.2466
D2	16.93	30	49	137.15	140.82	0.2187
D5-20	18.02	30	57	130.58	116.08	0.2297

## Data Availability

Data are contained within the article.

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
