# Peer review of "Mechanical, Flame-Retardant and Dielectric Properties of Intumescent Flame Retardant/Glass Fiber-Reinforced Polypropylene through a Novel Dispersed Distribution Mode"

_polymers, 2024, doi:10.3390/polym16101341_

Round 1

Reviewer 1 Report

Comments and Suggestions for Authors

In this study by Sun and co-worker, dispersed distribution modes of IFR/GF/PP were investigated to pursue a good flame-retardant property while maintain reasonably good mechanical and dielectric properties. However, there are some small issues need to address. A minor revision is recommended before publication.

1. In the introduction, a better presentation (a figure or a table) to easily compare previous studies and current study is appreciated.

2. More introduction about mechanical and dielectric properties is needed to better understand the corresponding test.

3. In the conclusion part, the first two paragraph are a little confusing. “Reducing the mechanical properties (line 396)” and “obtain better mechanical properties (line 402-403)” seems have opposite meanings.  It is better to summarize the pros and cons of different modes in an easily understandable manner (for example using a table). 

Author Response

Thank you very much for reviewing this article on your busy schedule. Your pertinent comments are very important to me and I have made the following changes in response to your comments:

  1. The schematic diagrams of four distribution modes of flame retardants were changed(Fig.1.), and the changed diagrams better reflect the problems of previous studies.
  2. Expanded the content of the mechanical and dielectric properties test methods, adding the specimen size, span-to-thickness ratio, the number of specimens and other relevant content. More details are provided in section 2.5.
  3. 396 is that for the same flame retardant level, uniform distribution requires more flame retardant, and more flame retardant results in lower mechanical properties and higher dielectric constant.402-403 is that when the flame retardant content is the same, the mechanical properties of uniform distribution are better, but the flame retardant properties of uniform distribution at this point are poorer than non-uniform distribution. Statements have been modified to make the intent of this paper clearer. More details are provided in section 4.

Reviewer 2 Report

Comments and Suggestions for Authors

Overall, the manuscript presents a comprehensive study on the mechanical, dielectric, and flame retardant properties of composite laminates with flame retardants distributed in different modes. The experimental setup, results, and conclusions are well-structured and provide valuable insights into the effects of distribution modes on the properties of the composite laminates.

Queries to the Authors:

1.      Can you provide more details on the experimental setup used for fabricating the composite laminates? Specifically, how were the flame retardants distributed in each distribution mode, and what techniques were employed to ensure uniformity?

2.      Have the mechanical, dielectric, and flame retardant properties been validated using multiple samples and experimental runs to ensure the reliability and reproducibility of the findings?

3.      What are the limitations of the study, and how might they affect the interpretation of the results? For example, are there any external factors or parameters that were not accounted for in the experimental design?

4.      How do the results of this study compare with existing literature on similar topics? Are there any discrepancies or inconsistencies that need to be addressed?

5.      Can you provide insights into the mechanisms underlying the observed changes in mechanical, dielectric, and flame retardant properties with different distribution modes? Are there any theoretical models or hypotheses that could explain these phenomena?

6.      Based on the findings, are there any optimization strategies that can be proposed for enhancing the overall performance of the composite laminates? For example, can certain distribution modes be favored for specific applications?

7.      What are the practical implications of the study findings for industries or applications where these composite laminates might be used? How feasible are the proposed distribution modes in real-world manufacturing processes?

8.      The manuscript discusses how mechanical properties vary with flame retardant content and distribution mode. Could you elaborate on the interplay between flame retardant content and specific mechanical properties such as bending strength and modulus? Are there any optimal flame retardant concentrations that maximize mechanical performance in each distribution mode?

9.      The study indicates a correlation between flame retardant content and dielectric properties. Can you provide insights into the mechanisms driving the observed variations in dielectric constant and loss tangent with different distribution modes? Additionally, how do these variations influence the suitability of the composite laminates for specific electronic applications?

10.  The manuscript highlights the effects of distribution mode on flame retardant properties. Could you further discuss how the dispersion of flame retardants influences flame propagation and self-extinguishing behavior? Additionally, are there any insights into how distribution mode affects the formation and effectiveness of carbonaceous char layers during combustion?

11.  The study compares gradient and dispersed distribution modes for flame retardants. Can you provide a more detailed comparative analysis of these modes, focusing on their respective advantages and limitations in enhancing mechanical, dielectric, and flame retardant properties? Furthermore, are there specific application scenarios where one distribution mode might be preferred over the other?

Add the following to enhance the quality of the manuscript.

12.  Provide a detailed explanation of the flame retardant loading levels used and their relevance to real-world applications.

13.  Delve deeper into the underlying mechanisms influencing mechanical properties, such as interface interactions and composite microstructure.

14.  Offer additional analysis on factors affecting dielectric behavior, such as filler morphology and frequency dependence.

15.  Provide practical suggestions for optimizing composite formulations and addressing manufacturing challenges based on the study findings.

Comments on the Quality of English Language

Minor grammar check is suggested. 

Author Response

Thank you very much for reviewing this article on your busy schedule. Your questions are very pertinent and the author have given them considerable thought and attempted to answer them in detail. Here are the author's responses to each question.

  1. In the preparation of IFR/PP pellets, in order to ensure that the flame retardant and PP are mixed uniformly, the extruding and pelletizing is repeated three times after mixing and pelletizing. In the preparation of IFR/GF/PP prepreg, IFR/PP pellets are melted through the extruder, and IFR/PP is compounded with continuous fiber in the head compounding die, and there is a corrugated structure in the compounding die, so that the resin fully infiltrates the fiber to ensure that the resin and fiber are uniformly compounded. A detailed schematic diagram of the preparation process has been added to the article(Fig.2).
  1. Mechanics (bending properties) are five parallel specimens, the data given in the text are average values and the error bars are standard deviations. Flame retardant oxygen index is about 15 specimens, flame retardant grade is 10 specimens, the experimental results are statistical results. Considering the price of the test, the dielectric test is four specimens, two each in theta=0°and theta=90°, the parallel specimen results do not differ much, and the trend of the test results in the two directions is the same, so the experiment is considered to be reliable and repeatable. The above has been added to the corresponding content of subsection 2.5.
  1. The thickness of a single layer of prepreg is a fixed value in this paper and is a factor that is not taken into account. But it can actually be varied. However, it is difficult to produce thinner prepreg layers in the laboratory's own small production line, and it is believed that better performance can be obtained if low or no flame retardant layers are dispersed in thinner layer thicknesses.
  2. The uniform and gradient distributions in the paper are the distribution modes proposed by the previous authors, and the dispersed distribution in the paper is proposed by the authors for the first time. In this paper, experimental comparisons are made between the three distribution modes to determine that the dispersed distribution proposed by the authors has a positive effect on the overall performance of the composite laminates and is superior to the uniform and gradient distributions proposed by the previous authors. The authors believe that the area that needs to be solved in this paper which is in agreement with previous studies is the problem of flame burning along the low flame retardant layer leading to cracking of the laminates. This issue could be the subject of subsequent research.
  1. That it is necessary to provide insights into the mechanisms underlying the observed changes in mechanical, dielectric, and flame retardant properties with different distribution modes, the authors add relevant content(section 4). The formula for calculating the flexural modulus of the laminate is given, and the flexural modulus of the G3 and D3 laminates is calculated. The results of this study provide theoretical support for the better bending properties of the flame retardant-free layers with excellent mechanical properties when it is close to the surface layer. A schematic diagram of the burning mechanism of gradient distribution and dispersed distribution is given, and the reason why the dispersed distribution has good flame retardant properties is explained. The theoretical equation between flame retardant content and dielectric properties is given, and the relationship between flame retardant content and dielectric properties is explained.
  1. On the premise of satisfying the flame retardant property, placing the low flame retardant layer with excellent mechanical properties closer to the surface layer can obtain better bending property. In this paper, it is considered that the dispersion distribution is more advantageous in the case of low dielectric excellent mechanical properties and flame retardant properties.
  1. The results of this study can provide ideas for the placement of high flame retardant layers in laminates. For example, it was placed evenly or all on the outside, which would reduce the mechanical properties, while the dispersed distribution not only had good flame retardant properties but also had high mechanical properties.
    Thermoplastic composite prepregs are in the form of sheets, and thermoplastic composite preparation is usually to cut the prepreg into the desired shape and size, stack them layer by layer, and then heating and pressurizing.For the manufacture of IFR/GF/PP in different distribution modes, it is only necessary to lay IFR/GF/PP prepregs with different flame retardant contents in the design order when laying prepregs (as Fig. 2).
  2. Mechanical properties are in conflict with flame retardant properties and need to be given balanced values depending on the application.
  3. In this study, the dielectric properties are mainly related to the content of flame retardants, and the increasing trend with the increase of flame retardant content,but has little relationship with the distribution of flame retardants. Signal transmission is related to the dielectric properties of the material, and low dielectric properties can reduce transmission losses and increase transmission rates. The dispersed distribution proposed in this study reduces the dielectric properties of IFR/GF/PP laminates by reducing the flame retardant content, so as to reduce the transmission loss and increase the transmission rate.
  4. After the high flame retardant layer is burned, the expanded carbon layer is generated, blocking the channel between the low flame retardant layer or the non-flame retardant layer and the flame, so as to achieve the flame retardant effect. This can be explained by Fig.
  5. In the case of bending load, the dispersed distribution is better than the gradient distribution. This part is added to the section 3.4 by theoretical analysis.

Questions 12 and 13 were not considered due to the fact that a large number of studies have been done on the interfacial effects of flame retardant additives to resins or fiber-reinforced composites. The focus of this paper is on the mode of macroscopic distribution of flame retardants

  1. The influence of filler morphology on dielectric properties has been studied by previous researchers(Zhao X, Li X, Lin X, et al. Effect of the microscopic morphology of a filler on its dispersion in a PI matrix: Calculation and analysis[J]. Journal of Applied Polymer Science. 2018, 136(7): 46875.) . The IFR used in this paper is a single type and a single form, and it is not a trace addition, so the author believes that the influence of the microstructure on the overall dielectric properties under the IFR addition in this paper can be ignored. The dielectric properties tend to decrease with increasing frequency. This is determined by the time of polarization. There is a brief mention in the text.
  1. This question is refined at the conclusion.